# Effects of Royal Jelly on Gut Dysbiosis and NAFLD in *db*/*db* Mice

**DOI:** 10.3390/nu15112580

**Published:** 2023-05-31

**Authors:** Genki Kobayashi, Takuro Okamura, Saori Majima, Takafumi Senmaru, Hiroshi Okada, Emi Ushigome, Naoko Nakanishi, Yuichiro Nishimoto, Takuji Yamada, Hideto Okamoto, Nobuaki Okumura, Ryoichi Sasano, Masahide Hamaguchi, Michiaki Fukui

**Affiliations:** 1Department of Endocrinology and Metabolism, Graduate School of Medical Science, Kyoto Prefectural University of Medicine, Kyoto 602-8566, Japan; genkoba@koto.kpu-m.ac.jp (G.K.); d04sm012@koto.kpu-m.ac.jp (T.O.); saori-m@koto.kpu-m.ac.jp (S.M.); semmarut@koto.kpu-m.ac.jp (T.S.); conti@koto.kpu-m.ac.jp (H.O.); emis@koto.kpu-m.ac.jp (E.U.); naoko-n@koto.kpu-m.ac.jp (N.N.); michiaki@koto.kpu-m.ac.jp (M.F.); 2Metabologenomics Inc., Tsuruoka 997-0052, Japan; yuichiro.nishimoto@metagen.co.jp (Y.N.); yamada@metagen.co.jp (T.Y.); 3Department of Life Science and Technology, Tokyo Institute of Technology, Tokyo 152-8550, Japan; 4Institute for Health Science, R&D Department, Yamada Bee Company, Inc., Okayama 708-0393, Japan; ho1993@yamada-bee.com (H.O.); no1780@yamada-bee.com (N.O.); 5AiSTI SCIENCE Co., Ltd., Wakayama 640-0033, Japan; sasano@aisti.co.jp

**Keywords:** royal jelly, non-alcoholic fatty liver disease, medium-chain fatty acids, dysbiosis, gut microbiota

## Abstract

Royal jelly (RJ) is a naturally occurring substance synthesized by honeybees and has various health benefits. Herein, we focused on the medium-chain fatty acids (MCFAs) unique to RJ and evaluated their therapeutic efficacy in treating non-alcoholic fatty liver disease (NAFLD). We examined *db*/*m* mice that were exclusively fed a normal diet, *db*/*db* mice exclusively fed a normal diet, and *db*/*db* mice fed varying RJ quantities (0.2, 1, and 5%). RJ improved NAFLD activity scores and decreased gene expression related to fatty acid metabolism, fibrosis, and inflammation in the liver. RJ regulated innate immunity-related inflammatory responses in the small intestine and decreased the expression of genes associated with inflammation and nutrient absorption transporters. RJ increased the number of operational taxonomic units, the abundance of *Bacteroides*, and seven taxa, including bacteria that produce short-chain fatty acids. RJ increased the concentrations of RJ-related MCFAs (10-hidroxy-2-decenoic acid, 10-hydroxydecanoic acid, 2-decenedioic acid, and sebacic acid) in the serum and liver. These RJ-related MCFAs decreased saturated fatty acid deposition in HepG2 cells and decreased the gene expression associated with fibrosis and fatty acid metabolism. RJ and RJ-related MCFAs improved dysbiosis and regulated the expression of inflammation-, fibrosis-, and nutrient absorption transporter-related genes, thereby preventing NAFLD.

## 1. Introduction

The occurrence of non-alcoholic fatty liver disease (NAFLD) has been progressively increasing, particularly in developed nations, parallel to the rising prevalence of obesity and diabetes [1,2]. Overeating and a sedentary lifestyle lead to the onset of NAFLD; thus, metabolic diseases, such as obesity, diabetes, and dyslipidemia, often coexist with NAFLD in many patients [3,4]. Patients with NAFLD have a higher mortality rate compared to those without NAFLD [5]; thus, the prevention of NAFLD is an issue of utmost importance.

Dysbiosis, referring to the imbalance or disruption of the gut microbiota, is closely related to the incidence of NAFLD [6]. This condition is characterized by a reduction or loss of some beneficial commensal microorganisms and an increase in potentially pathogenic commensal microorganisms and can arise from various causes, such as dietary choices, food additives, antibiotics, and lifestyle [7]. Furthermore, dysbiosis is associated with the dysfunction of the intestinal barrier and hepatic inflammation due to infiltration of the liver by microbial or food antigens via the portal vein [6].

Royal jelly (RJ) is a naturally occurring substance synthesized by honeybees and fed to both the queen bee and young worker larvae [8,9]. RJ contains three major nutrients, namely proteins (including abundant essential amino acids), carbohydrates, and fats, and is a source of various vitamins and minerals. RJ plays a vital role in promoting the optimal growth of queen honeybees and is closely linked to their reproductive capabilities and lifespan. Research has provided insights into the fact that RJ is a functional food that possesses diverse properties beneficial for human health. For instance, RJ has been reported to have various effects on metabolic diseases such as glucose intolerance, lipid disorders, and hypertension [10,11,12,13]. Notably, RJ and major RJ proteins have been reported to improve NAFLD [14,15,16]. Four main types of medium-chain fatty acids (MCFAs) have been identified in RJ: 10-hydroxy-2-decenoic acid (10 H2DA), 10-hydroxydecanoic acid (10 HDAA), 2-decenedioic acid (2-DA), and sebacic acid (SA). 10 HDAA and SA are metabolites of 10 H2DA and 2-DA, respectively [17]. However, the anti-NAFLD effects of these MCFAs are still unclear.

This study focused on the MCFAs present in RJ and dysbiosis. We attempted to elucidate the mechanism by which RJ improves NAFLD in *db*/*db* mice with severe NAFLD by measuring the tissue concentrations of RJ-related MCFAs and evaluating intestinal tissues, small intestinal inflammation, gene expression in the intestine, and gut microbiota dysbiosis.

## 2. Materials and Methods

### 2.1. Mice

We purchased 7-week-old male diabetic homozygous *db*/*db* mice and non-diabetic heterozygous *db*/*m* mice from Shimizu Laboratory Supplies (Kyoto, Japan). We used *db*/*db* mice because they exhibit overeating, obesity, hyperglycemia, and NAFLD owing to leptin receptor abnormalities. Female mice have altered innate immunity as a result of their sex hormone cycle. Male mice, on the other hand, have less altered innate immunity due to the sex hormone cycle, so we employed male mice [18]. Therefore, we used male mice in our study. The RJ powder used in this study was derived from RJ treated with alkaline proteases to eliminate allergens and standardized to contain 3.5% 10 H2DA and 0.6% 10 HDAA (lot: YRP-M-210728-1; Yamada Bee Company, Inc., Okayama, Japan) [19]. Mice (8-week-old) were fed either a normal diet (ND; 344.9 kcal/100 g, fat kcal 4.6%; Oriental Yeast Japan, Tokyo, Japan) or ND supplemented with RJ (0.2, 1, or 5% *w*/*w*) for 8 weeks. The mice were categorized into five groups (*n* = 6 each): (1) *db*/*m* mice fed a diet without RJ (*db*/*m*), (2) *db*/*db* mice fed a diet without RJ (*db*/*db*), (3) *db*/*db* mice fed an ND supplemented with 0.2% RJ (*db*/*db* + 0.2% RJ), (4) *db*/*db* mice fed an ND supplemented with 1% RJ (*db*/*db* + 1% RJ), and (5) *db*/*db* mice fed an ND supplemented with 5% RJ (*db*/*db* + 5% RJ) (Figure 1A). We supplied equal amounts of feed to both *db*/*m* and *db*/*db* mice and performed paired feeding. After an overnight fasting period, euthanasia was performed on all mice at 16 weeks of age using a combination of anesthetics, including 5.0 mg/kg of butorphanol, 0.3 mg/kg of medetomidine, and 4.0 mg/kg of midazolam. All animal experiments were approved by the Committee for Animal Research, Kyoto Prefectural University of Medicine, Kyoto, Japan (approval number: M2020-48).

### 2.2. Momentum Measurement

The mice were kept individually in cages with a running wheel (MK-713; Muromachi Kikai, Tokyo, Japan). Before being euthanized, we recorded the count of rotations made by the running wheel in each individual cage every 12-h night cycle for 5 days. We used CompACT AMS ver. 3 software (Muromachi Kikai), which was connected to the running wheels to record the number of rotations.

### 2.3. Glucose and Insulin Tolerance Tests

Due to the association of dysbiosis with inflammation, NAFLD, diabetes, and insulin resistance, we conducted glucose and insulin tolerance tests. At 15 weeks of age, mice underwent intraperitoneal glucose tolerance testing (iPGTT) with a dose of 2 g/kg body weight after a fasting period of 16 h. Additionally, they underwent insulin tolerance testing (ITT) with a dose of 0.5 U/kg body weight following a fasting period of 5 h. These tests were performed as described previously [20].

### 2.4. Biochemical Analysis

Fasting mice were used to collect blood samples, and the levels of alanine aminotransferase (ALT), aspartate aminotransferase (AST), total cholesterol (T-Chol), triglycerides (TG), and non-esterified fatty acids (NEFA) were evaluated. Biochemical analysis was conducted at Fujifilm Wako Pure Chemical Corporation (Osaka, Japan).

### 2.5. Liver Histological Analysis

Liver tissues were taken from all mice and subjected to Masson’s trichrome and hematoxylin and eosin (HE) staining. The tissues were immersed in a 10% buffered formaldehyde solution for fixation. For the purpose of oil red O staining, liver tissues were additionally subjected to fixation in a 4% paraformaldehyde phosphate buffer solution. The severity of NAFLD and fibrosis stage were determined as per the NAFLD activity score [21]. Experiments were performed as described previously [22].

### 2.6. Isolation of Mononuclear Cells from the Liver and the Small Intestine

Before harvesting or washing the tissues with phosphate-buffered salts (PBS), systemic perfusion was performed using heparinized saline to minimize the risk of blood contamination. Experiments were performed as described previously [23].

### 2.7. Histological Analysis of the Small and Large Intestines

The colon and jejunum were excised from the mice and promptly immersed in 10% buffered formaldehyde at 22 °C for 24 h. Histological analysis was described as described previously [22,24,25].

### 2.8. Flow Cytometry

The analysis of stained cells was conducted using a FACS Canto II system, and the resulting data were evaluated by FlowJo ver. 10 software (TreeStar, Ashland, OR, USA). The evaluation procedures were carried out following previously described methods. Analyses were performed as described previously [23].

### 2.9. Gene Expression Analysis

Following a 16-h fasting period, the livers and small intestines of the mice were removed and promptly frozen using liquid nitrogen. Gene expression analysis was performed as described previously [20].

### 2.10. 16S rRNA Sequencing

For the bacterial flora analysis, the 16S rRNA could not be classified in detail at the species level, so the analysis was performed at the genus level. To conduct 16S rRNA sequencing, we adhered to established protocols [23,24,25]. Frozen appendicular fecal samples were used, and microbial DNA was extracted using the QIAamp DNA Feces Mini Kit (Qiagen, Venlo, The Netherlands), following the protocol outlined by the manufacturer. A bacterial universal primer set (341F and 806R) was utilized to amplify the V3–V4 region of the 16S rRNA gene from the DNA samples. PCR was conducted using EF-Taq (Solgent, Korea) with a 30 µL reaction mixture that included 20 ng of genomic DNA as the template. The thermocycling parameters consisted of an initial activation step at 95 °C for 2 min, followed by 35 cycles at 95 °C, 55 °C, and 72 °C for 1 min each, and a final step at 72 °C for 10 min. The amplified products underwent purification using a multiscreen filter plate (Millipore Corp., Billerica, MA, USA). For 16S rRNA sequencing, a MiSeq sequencer (Illumina, CA, USA) was utilized following the protocol outlined by the manufacturer (Macrogen, Seoul, Korea). QIIME version 1.9 was employed for quality filtering of the sequences, and barcodes or primers with scores below 75% were excluded from the analysis. The number of operational taxonomic units (OTUs) was calculated utilizing the UCLUST algorithm with a similarity threshold of 97%. Additionally, for the taxonomic classification of 16S rRNAs, BLAST (UNITE, 2017) was employed with the UNITE sequence set of the Greengenes core set aligned with UCLUST and ITS.

The Kyoto Encyclopedia of Genes and Genomes (KEGG) ortholog abundance predictions were obtained using the Phylogenetic Investigation of Communities by Reconstruction of Unobserved States (PICRUSt2) software (version 2.4.1).

The relative abundance of phyla within the groups was assessed using a one-way ANOVA with the Holm–Šídák multiple-comparison test. To analyze the alpha diversity, which refers to the diversity within individual samples, we employed the Chao1, Shannon, and Gini–Simpson indices.

We employed linear discriminant analysis (LDA) coupled with effect size measurements (LEfSe) to assess the relative abundance of bacterial genera among the groups (available online: http://huttenhower.sph.harvard.edu/lefse/ (accessed on 15 May 2022).

Using a normalized relative abundance matrix, LEfSe detected taxa exhibiting statistically significant differences in abundance, and the effect size of each feature was assessed using LDA. A significance threshold of 0.05 (Wilcoxon rank-sum test) and an effect size threshold of 2 were employed for all biomarkers examined.

Furthermore, we conducted a principal component analysis (PCA) to evaluate the effectiveness of RJ. Additionally, nonhierarchical K-means cluster analysis was performed, with the number of clusters to be generated prespecified as 2, using Tinn-R Gui version 1.19.4.7 and R version 1.36.

### 2.11. Measurement of Palmitic Acid, Short-Chain Fatty Acid (SCFA), and Middle-Chain Fatty Acid Concentrations

The levels of palmitic acid in the serum, liver, and feces, as well as SCFAs in the serum and feces and MCFAs in the serum and liver, were determined by gas chromatography-mass spectrometry (GC/MS) using an Agilent 7890B/7000D system (Agilent Technologies, Santa Clara, CA, USA). Measurements were conducted according to the previously outlined procedure [23].

### 2.12. Human Hepatoma Cell Culture

The HepG2 cell line was obtained from KAC Company (Kyoto, Japan). The cells were incubated in a minimum essential medium supplemented with 10% heat-inactivated fetal bovine serum, 1% penicillin–streptomycin, 2 mM l-glutamine, and 1 mM sodium pyruvate. To maintain the cells, they were cultured in a humidified incubator with 5% CO2 at 37 °C. Twenty-four to forty-eight hours prior to treatments, cells were seeded in a 96-well plate and allowed to reach 70% confluence. Twenty-four hours after changing the medium, the cells underwent treatment with ethanol (Ctrl), 200 μM palmitic acid (PA), or 200 μM PA and 100 μg/mL of 10 H2DA (PA + 10 H2DA), 100 μg/mL of 10 HDAA (PA + 10 HDAA), 100 μg/mL of 2-decenoic acid (PA + 2-DA), or 100 μg/mL of SA (PA + SA) (Day 1). We performed oil red O staining and real-time reverse transcription-polymerase chain reaction (RT-PCR) 24 h following treatment (day 2).

### 2.13. Oil Red O Staining

HepG2 cells were subjected to 30-min fixation in 4% paraformaldehyde, followed by PBS washing. Experiments were performed as described previously [18].

### 2.14. Gene Expression Analysis of HepG2 Cells

RT-PCR was performed to assess gene expression. Experiments were performed as described previously [18].

### 2.15. Statistical Analysis

The data underwent analysis utilizing JMP ver. 13.0 software. (SAS, Cary, NC, USA). To compare the results among different groups, one-way analysis of variance and Holm-Šídák’s multiple comparisons test were employed. An unpaired *t*-test was utilized to compare the results between the two groups. We determined statistical significance at a threshold of *p* < 0.05. GraphPad Prism ver. 9.0 software (San Diego, CA, USA) was utilized for generating figures.

## 3. Results

### 3.1. Administration of RJ Improved Locomotor Activity, Glucose Metabolism, and Lipid Metabolism

The body weight of *db*/*db* mice was higher compared with RJ-fed *db*/*db* mice (Figure 1B). Locomotor activity in RJ-fed *db*/*db* mice was higher than that in *db*/*db* mice (Figure 1C). Moreover, iPGTT and ITT revealed that RJ improved glucose tolerance and insulin sensitivity (Figure 1D). Serum ALT, T-chol, TG, and NEFA levels in *db*/*db* mice were higher than those in *db*/*db* mice fed RJ (Figure 1E,F).

### 3.2. Administration of RJ Improved NAFLD

The absolute liver weight in *db*/*db* mice was lower than that in RJ-fed mice, whereas the differences in relative liver weights between the two groups were not significant (Figure 2A). The NAFLD activity scores, which were utilized to evaluate the severity of NAFLD and stage of fibrosis, in *db*/*db* mice were higher than those in RJ-fed *db*/*db* mice (Figure 2B). Additionally, fat deposition in the liver of *db*/*db* mice was greater than that in RJ-fed mice (Figure 2C,D).

### 3.3. Administration of RJ Improved Visceral Fat, Obesity, and Skeletal Muscle Loss

The epididymal fat weight of *db*/*db* mice and epididymal fat weight/BW of *db*/*db* mice were higher than those of RJ-fed *db*/*db* mice (Appendix A). Histological images of epididymal fat tissues are shown in Appendix A. The adipocyte size in *db*/*db* mice was larger than that in RJ-fed *db*/*db* mice (Appendix A). In addition, the soleus muscle weight of *db*/*db* mice was lower than that of RJ-fed *db*/*db* mice (Appendix A).

### 3.4. Administration of RJ Improved Inflammation and Decreased the Expression of Genes Related to Fatty Acid Metabolism, Inflammation, and Fibrosis in the Liver

The ratios of the innate lymphoid cell (ILC) 1 or ILC3, which are associated with inflammation, to CD45-positive cells in *db*/*db* mice, were higher than those in *db*/*m* mice and *db*/*db* mice fed RJ. The ratio of M1 macrophages (pro-inflammatory) to M2 macrophages (anti-inflammatory) in *db*/*db* mice was higher than that in *db*/*m* mice and *db*/*db* mice fed RJ (Figure 2E). The relative expression of genes associated with fatty acid metabolism-related enzymes (*Fasn* and *Scd1*) and inflammation and fibrosis (*Tnfa*, *ll1b*, *Ccl2*, *Ifng*, and *Col1a*) in *db*/*db* mice was higher than that in *db*/*m* mice and *db*/*db* mice fed 5% RJ (Figure 2F).

### 3.5. Administration of RJ Improved Atrophy of the Small and Large Intestinal Mucosa

Figure 3A displays the histological images of the jejunum and colon. Compared with *db*/*m* mice and *db*/*db* mice fed RJ, *db*/*db* mice exhibited lower villus height and width and atrophic jejunal mucosa. In contrast, the crypt depth/villus height ratio in *db*/*db* mice was notably higher than that in the other four groups. The number of goblet cells/crypts in *db*/*db* mice was lower than that in *db*/*m* mice and *db*/*db* mice fed RJ (Figure 3B).

### 3.6. Administration of RJ Regulated Inflammatory Responses in Innate Immunity in the Small Intestine

In the small intestine, ILC3 cells produce antimicrobial peptides. Ex-ILC3 cells are associated with inflammation by acting similarly to ICL1 cells. The ratio of ILC1 and ex-ILC3 cells to CD45-positive cells in the small intestine of *db*/*db* mice was higher than that in *db*/*m* mice and *db*/*db* mice fed RJ. The ratio of ILC3 cells to CD45-positive cells in *db*/*db* mice was lower than that in *db*/*m* mice and *db*/*db* mice fed RJ. Moreover, the ratio of M1 macrophages to M2 macrophages in *db*/*db* mice was higher than that in *db*/*m* mice and *db*/*db* mice fed RJ (Figure 3C).

### 3.7. Administration of RJ Increased the Expression of Genes Related to Mucin Secretion in the Small Intestine and Decreased the Expression of Genes Related to Inflammation and Nutrient Absorption Transporters

Il22 is associated with enhanced mucin secretion, which acts as a barrier and supports the symbiosis between the human body and gut microbiota [26]. The relative expression of *Il22* was lower in *db*/*db* mice than in *db*/*m* mice, whereas RJ increased the expression of this gene. The expression of inflammation-related genes (*Tnfa* and *Ifng*) and nutrient transporter genes (*Cd36*, *Sglt1*, and *Pept1*) was higher in *db*/*db* mice than that in *db*/*m* mice; however, this expression was significantly decreased upon RJ administration (Figure 3D).

### 3.8. Administration of RJ Improved Dysbiosis

OTUs in *db*/*db* mice were lower than those in *db*/*m* mice and *db*/*db* mice fed RJ. The Chao1 and Shannon indices, which are indicators of the α-diversity of gut microbiota, were lower in *db*/*db* mice than those in *db*/*m* mice. However, these indices improved after RJ administration (Figure 4A). In *db*/*db* mice, the most abundant phylum was Firmicutes, which is associated with diabetes and obesity, and in *db*/*m* and *db*/*db* mice fed 5% RJ, the most abundant phylum was Bacteroidetes (Figure 4B). Seven taxa, including Firmicutes, were observed to be overexpressed in *db*/*db* mice, whereas seven taxa, including SCFA-producing bacteria, such as the family *Ruminococcaceae*, genus *Butyricicocus*, and genus *Acetivibrio*, were overexpressed in *db*/*db* mice fed 5% RJ (Figure 4C). In addition, PCA showed that *db*/*m* and *db*/*db* mice fed 5% RJ belonged to the same cluster (Figure 4D).

### 3.9. Administration of RJ Increased Palmitic acid Excretion and SCFA Production in Feces and Increased MCFAs in the Blood and Liver

Next, we measured the concentration of palmitic acid in the feces, serum, and liver. Palmitic acid concentrations in the liver and serum of *db*/*db* mice were higher than those in *db*/*m* mice, and these levels decreased upon administration of RJ. In contrast, the concentration of palmitic acid in the feces was lower in *db*/*db* mice than that in *db*/*m* mice, and these levels increased upon RJ administration (Figure 5A). Levels of SCFAs, such as acetic, butyric, and propionic acids, were significantly higher in the feces and serum of *db*/*db* mice fed RJ than those observed for *db*/*db* mice (Figure 5B,C). We also measured the concentrations of RJ-specific MCFAs, such as 10H2DA, 10HDAA, SA, and 2-DA, in the serum and liver of mice (Figure 5D,E). RJ increased the concentration of MCFAs in *db*/*db* mice.

### 3.10. RJ-Related MCFAs Decreased Saturated Fatty Acid Deposition in HepG2 Cells and Decreased the Expression of Genes Related to Fatty Acid Metabolism and Fibrosis

We treated HepG2 cells with palmitic acid and RJ-related MCFAs (10H2DA, 10HDAA, SA, or 2-DA) and analyzed the cells for fat deposition and gene expression. The oil red O-positive area in HepG2 cells corresponding to RJ-related MCFAs was smaller than that with palmitic acid alone (Figure 6A). The relative expression of *FASN* and *SCD1*, which are genes associated with fatty acid metabolism, and *COLA1A*, which is a gene associated with fibrosis, was lower in the presence of RJ-related MCFAs than in the presence of only palmitic acid (Figure 6B–D). In particular, the relative expression of these genes decreased following treatment with 10H2DA or SA (Appendix A).

## 4. Discussion

In this study, RJ improved dysbiosis and inflammation of the intestinal mucosa, increased saturated palmitic acid excretion in the feces and SCFA-producing bacteria in the intestine, decreased the concentration of palmitic acid in the blood and liver, and increased MCFAs in the liver, thereby resulting in an improvement in NAFLD. This study is the first to elucidate that RJ regulates the expression of nutrient absorption transporter genes in the small intestine in vivo, measure the concentration of RJ-related MCFAs in the liver, and show that RJ and RJ-related MCFAs may improve NAFLD.

In our study, RJ administration improved dysbiosis. In particular, SCFA-producing bacteria were overexpressed in the intestine of RJ-fed mice. SCFAs are used as a major source of nutrition by the mucosal cells in the large intestine and increase mucin expression in goblet cells, increase mucus production, and strengthen tight junctions between the epithelial cells of the intestine [20,27]. Impairment of the intestinal barrier results in systemic microinflammation due to lipopolysaccharides produced by gram-negative rods [28], and a previous study has demonstrated increased serum endotoxin levels in patients with non-alcoholic steatohepatitis [29]. In addition, SCFAs bind to the free fatty acid receptor and suppress the activation of nuclear factor-kappa B [30], resulting in the suppression of the secretion of cytokines such as IL1, tumor necrosis factor-α (TNF-α), and IL12 [31,32]. ILC1 cells produce TNF-α and interferon-gamma (IFN-γ) in response to IL12 [33], and ILC3 cells produce IL17 and IL22 in response to IL1β [34], resulting in cytotoxicity. In this study, RJ increased the excretion of palmitic acid in the feces and decreased its concentration in the serum and liver. SCFAs increase the secretion of GLP1, which slows the excretion of gastric contents [35] and delays nutrient absorption. High-fat diets have been reported to increase the expression of fat absorption transporters [36], and high-protein diets have been reported to increase the expression of Pept1, which is a protein absorption transporter found in the small intestine [37]. Other research has reported that high glucose conditions increase the expression of SGLT1 in the small intestine [38]. We envisage that the expression of nutrient absorption transporters in *db*/*db* mice increased because of overeating, whereas the administration of RJ decreased the expression of these transporters in the small intestine and increased GLP1, resulting in decreased nutrient absorption, including that of saturated fatty acids. RJ has been shown to improve serum lipid profiles in mice [16], which is consistent with our results. Saturated fatty acids in the diet have been reported to be related to increased IL12 secretion from macrophages, mitochondrial damage, chronic low-grade inflammation, metabolic disorders, and cardiovascular diseases [39,40,41,42]. Saturated fatty acids have also been implicated in muscle atrophy and progressive inflammation in the liver [20]. We envisage that the administration of RJ maintained intestinal barrier function, regulated inflammation, and decreased saturated fatty acid levels in the liver by improving gut microbiota dysbiosis, increasing SCFAs, and regulating the expression of nutrient absorption transporters in the small intestine. As a result, NAFLD improved. Further, the improvement in liver inflammation and decreased absorption of nutrients are thought to improve glucose intolerance, lipid metabolism, visceral fat accumulation, and muscle mass loss. In this study, RJ increased the liver size without causing fatty liver disease, possibly because RJ inhibited steatohepatitis in *db*/*db* mice because of the aforementioned effects. However, *db*/*db* mice are known to consume more food than *db*/*m* mice, resulting in fat deposition and hypertrophy of the liver and other organs. In addition, it is possible that the livers of *db*/*db* mice that were not fed RJ were atrophied because of inflammation and fibrosis.

In our study, the concentrations of 10H2DA, 10HDAA, 2-DA, and SA in the serum and liver increased after RJ administration. These RJ-related MCFAs decreased saturated fatty acid deposition and decreased gene expression related to fatty acid metabolism and fibrosis in HepG2 cells. MCFAs are absorbed through the portal system and quickly provide energy to the liver, thereby inhibiting hepatic fat accumulation as well as the development of dyslipidemia [16]. MCFAs have also been reported to activate energy metabolism by themselves, and their metabolites modify histone acetylation [43,44]. RJ contains several characteristic MCFAs. Particularly, 10H2DA, which is an unsaturated fatty acid unique to RJ, has been reported to have antimicrobial, anticancer, epidermal protective, lifespan-prolonging, and immunomodulatory effects [9,45,46,47,48,49,50]. RJ also contains 10HDAA (a saturated fatty acid of 10H2DA), 2-DA, and SA (a saturated fatty acid of 2-DA) [9]. 10HDAA has been reported to inhibit the synthesis of nitric oxide associated with inflammation [51]. Moreover, SA exerts a suppressive effect on inflammation by decreasing the level of TNF-α [52]. RJ-related MCFAs may regulate gene expression, improve liver inflammation, and contribute to the improvement of NAFLD. Previous studies have reported that major RJ proteins (MRJPs) improve obesity, fatty liver, and insulin resistance in mice, as well as fat accumulation in vitro [14,15]. However, MRJPs are known to get digested during the RJ powder production process [19]. A study showed that the concentrations of serum 2-DA and SA were increased by RJ or enzyme-treated RJ [17]. Our study revealed that among the components of RJ, i.e., not only MRJPs but also RJ-related MCFAs, some have anti-fat accumulation effects.

This study has a limitation. HepG2 cells have been used in this study, and we have not used primary hepatocytes. More precise experiments might have been possible if we had used primary hepatocytes.

## 5. Conclusions

RJ improved dysbiosis and regulated nutrient absorption-related gene expression in the small intestine, resulting in increased excretion of saturated fatty acids in feces, an increase in SCFA-producing intestinal bacteria, an improvement in inflammation in the intestine and liver, and an improvement in atrophy of the small and large intestinal mucosa. This study represents the initial evidence to demonstrate that RJ regulates the expression of nutrient absorption transporter genes in the small intestine in vivo, to measure RJ-related MCFA concentrations in the liver, and to show that RJ and RJ-related MCFA may improve NAFLD. Further clinical trials are required to validate our findings for the treatment of NAFLD.

## Figures and Tables

**Figure 1 nutrients-15-02580-f001:**
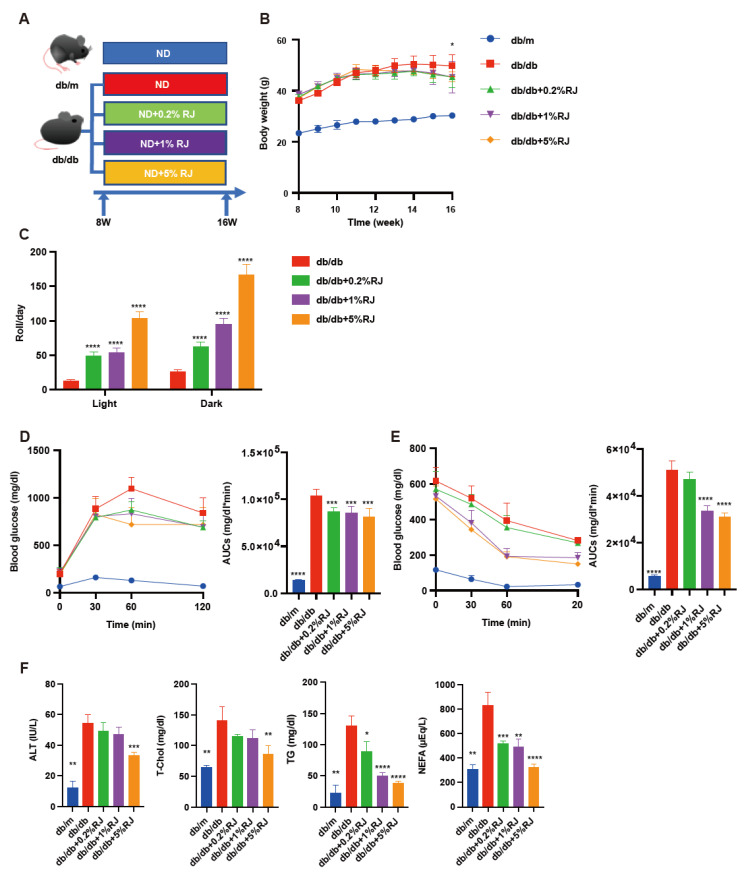
RJ-fed *db*/*db* mice showed higher locomotor activity, lower blood glucose levels, and lower liver enzymes and serum lipids compared with *db*/*db* mice that were not fed RJ. (**A**) Administration of RJ (0.2%, 1%, and 5% per feed weight) was initiated at 8 weeks of age. (**B**) Changes in the body weight (*n* = 6). (**C**) Spontaneous locomotor activity in light and dark phases (*n* = 6). (**D**) Results of intraperitoneal glucose tolerance test (2 g/kg body weight) for 15-week-old mice and AUC analysis (*n* = 6). (**E**) Results of insulin tolerance test (0.75 U/kg body weight) for 15-week-old mice and AUC analysis (*n* = 6). (**F**) Serum ALT, T-Chol, TG, and NEFA levels (*n* = 6). Data are presented as the mean ± standard deviation. Data were analyzed using a one-way analysis of variance and Holm–Šídák’s multiple comparison test. * *p* < 0.05, ** *p* < 0.01, *** *p* < 0.001, and **** *p* < 0.0001 compared to *db*/*db* mice. ALT, alanine aminotransferase; AUC, the area under the curve; NEFA, non-esterified fatty acid; RJ, royal jelly; T-Chol, total cholesterol; TG, triglycerides.

**Figure 2 nutrients-15-02580-f002:**
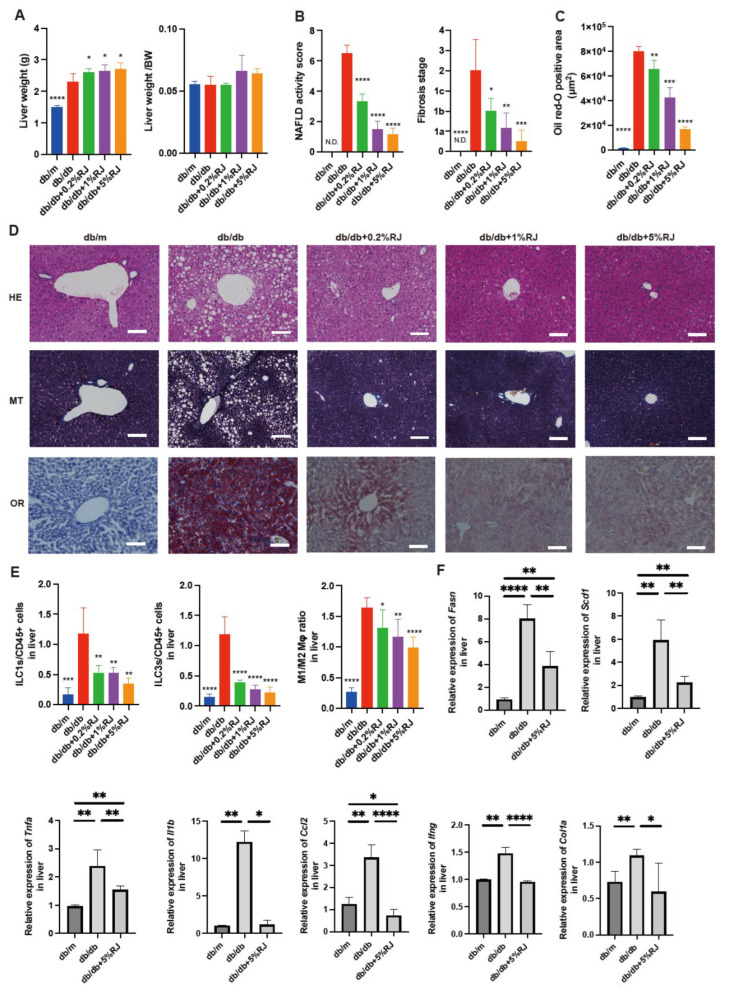
RJ-fed *db*/*db* mice had lower intrahepatic fat accumulation, inflammatory cells, and gene expression associated with inflammation compared with *db*/*db* mice that were not fed RJ. (**A**) Absolute and relative weights of the liver (*n* = 6). (**B**) NAFLD activity scores and the fibrosis stage (*n* = 6). (**C**) Oil red O staining positive area (μm^3^) (*n* = 6). (**D**) Representative images of HE-, MT-, and OR-stained liver sections. Liver tissues were obtained from 16-week-old mice. Scale bar = 100 μm. (**E**) The ratio of ILC1s, ILC3s, and M1 to M2 macrophages to CD45-positive cells in the liver (*n* = 6 in each case). (**F**) Relative *mRNA* expression of *Fasn*, *Scd1*, *Tnfa*, *Il1b*, *Ccl2*, *Ifng*, and *Col1a* in the liver, normalized to the expression of *Gapdh* (*n* = 6). Data are presented as the mean ± standard deviation. Data were analyzed using a one-way analysis of variance with Holm–Šídák’s multiple comparison test. * *p* < 0.05, ** *p* < 0.01, *** *p* < 0.001, and **** *p* < 0.0001 compared to *db*/*db* mice. HE, hematoxylin and eosin; MT, Masson trichrome; OR, oil red O; Mφ, macrophages; N.D., not-detected; NAFLD, non-alcoholic fatty liver disease; RJ, royal jelly; ILC, innate lymphoid cell.

**Figure 3 nutrients-15-02580-f003:**
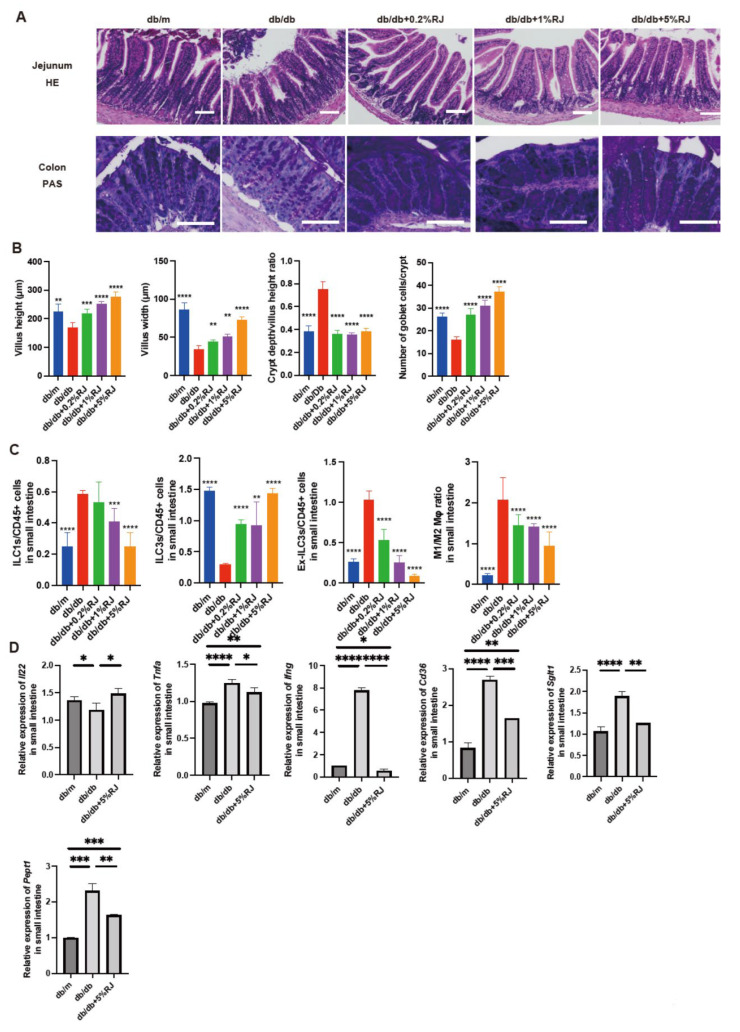
RJ-fed *db*/*db* mice showed lower intestinal inflammation compared with *db*/*db* mice that were not fed RJ. (**A**) Representative images of HE- and PAS-stained jejunum (HE) and colon (PAS) sections. Intestinal tissues were obtained from 16-week-old mice. (**B**) Villus height of the jejunum (μm), villus width of the jejunum (μm), crypt depth/villus height ratio, and the number of goblet cells/crypt. (**C**) The ratio of ILC1s, ILC3s, ex-ILC3s, and M1 to M2 macrophages to CD45-positive cells in the small intestine (*n* = 6 in each case). (**D**) Relative *mRNA* expression of *Il22*, *Tnfa*, *Ifng*, *Cd36*, *Sglt1*, and *Pept1* in the small intestine, normalized to the expression of *Gapdh* (*n* = 6). Data are presented as the mean ± standard deviation. Data were analyzed using a one-way analysis of variance and Holm–Šídák’s multiple comparison test. * *p* < 0.05, ** *p* < 0.01, *** *p* < 0.001, and **** *p* < 0.0001 compared to *db*/*db* mice in (**A**–**C**) and among three groups in (**D**). RJ, royal jelly; HE, hematoxylin and eosin; PAS, periodic acid Schiff; ILC, innate lymphoid cell.

**Figure 4 nutrients-15-02580-f004:**
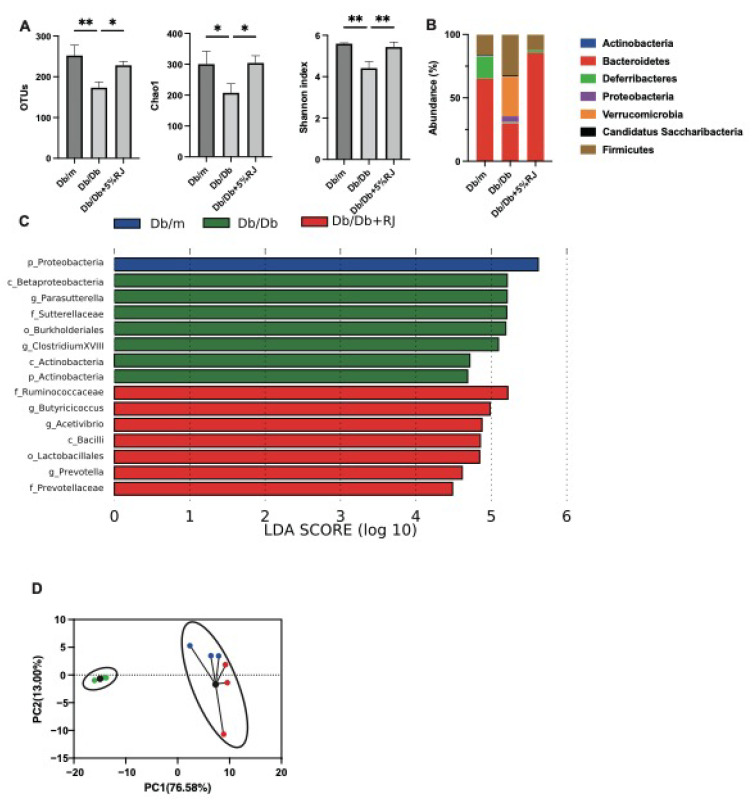
RJ-fed *db*/*db* mice showed a higher diversity of gut microbiota and a higher abundance of short-chain fatty acid-producing gut microbiota compared with *db*/*db* mice that were not fed RJ. (**A**) OTUs (*n* = 3), Chao1 index (*n* = 3), and Shannon index (*n* = 3). (**B**) Relative abundance of gut microbiota at the phylum level (*n* = 3). (**C**) LDA scores of gut microbiota of *db*/*m*, *db*/*db*, and *db*/*db* mice fed 5% RJ (*n* = 3). *db*/*m* mice (blue); *db*/*db* mice (green); *db*/*db* mice fed 5% RJ (red). (**D**) PCA and k-means clustering for gut microbiota (*n* = 3). Data were analyzed using a one-way analysis of variance and Holm–Šídák’s multiple comparison test. * *p* < 0.05 and ** *p* < 0.01 among the three groups. LDA, linear discriminant analysis; OTU, operational taxonomic units; PCA, principal component analysis.

**Figure 5 nutrients-15-02580-f005:**
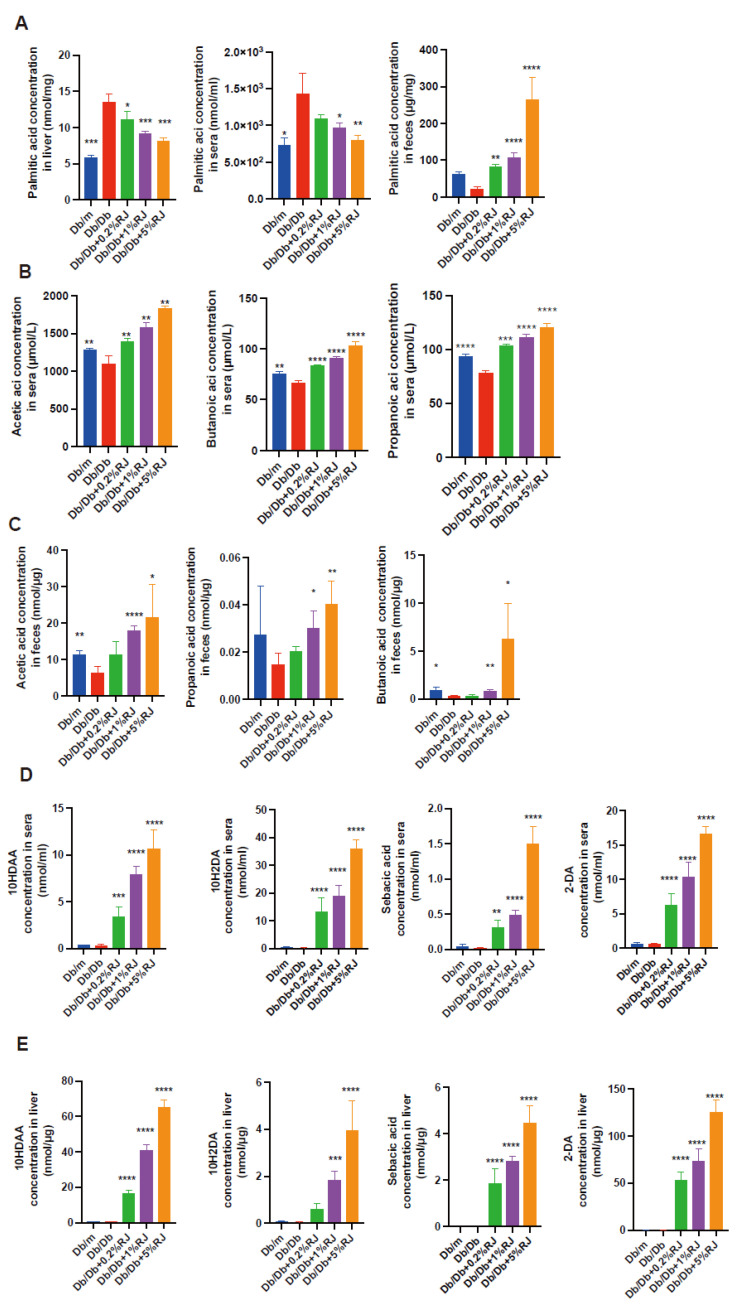
Metabolites in the liver, sera, and feces. (**A**) Concentrations of palmitic acid in the liver, sera, and feces (*n* = 6). (**B**,**C**) Concentrations of acetic acid, propanoic acid, and butanoic acid in the sera and feces (*n* = 6). (**D**,**E**) Concentrations of 10HDAA, 10H2DA, SA, and 2-DA in the sera and liver (*n* = 6). Data are presented as the mean ± standard deviation. Data were analyzed using a one-way analysis of variance and Holm–Šídák’s multiple comparison test. * *p* < 0.05, ** *p* < 0.01, *** *p* < 0.001, and **** *p* < 0.0001 compared to *db*/*db* mice. 10HDAA, 10-hydroxydecanoic acid; 10H2DA, 10-hidroxy-2-decenoic acid; 2-DA, SA, sebacic acid, 2-decenedioic acid.

**Figure 6 nutrients-15-02580-f006:**
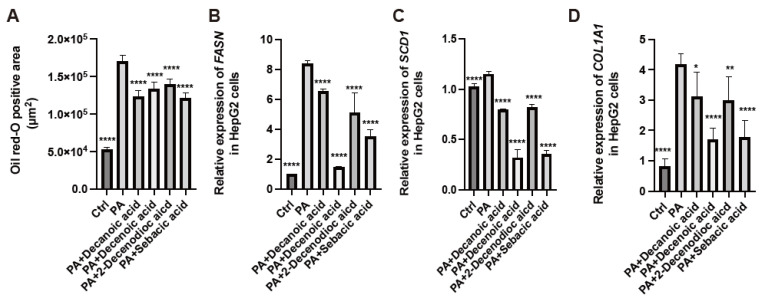
Administration of medium-chain fatty acids from royal jelly decreased the accumulation of saturated fatty acids and gene expression related to fatty acid synthesis and fibrosis in HepG2 cells. (**A**) Oil red-O positive area (*n* = 4). Relative mRNA expression of (**B**) FASN, (**C**) SCD1, and (**D**) COL1A1 in HepG2 cells treated with ethanol (Ctrl), 200 µM PA, 200 µM PA, and 0.5 mM 10 HDAA acid, 200 µM PA and 0.5 mM 10 H2DA acid, 200 µM PA and 0.5 mM 2-DA, and 200 µM PA and 0.5 mM SA (*n* = 4). Data are represented as the mean ± SD values. Data were analyzed using a 1-way ANOVA with Holm-Šídák’s multiple-comparisons test. * *p* < 0.05, ** *p* < 0.01, **** *p* < 0.0001 compared to *db*/*db* mice. PA, palmitic acid; 10 HDAA, 10-hydroxydecanoic acid; 10 H2DA, 10-hidroxy-2-decenoic acid; 2-DA, 2-decenedioic acid; SA, sebacic acid.

## Data Availability

The datasets of the study are available from the corresponding author upon reasonable request.

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
