# Peer review of "Effects of Royal Jelly on Gut Dysbiosis and NAFLD in db/db Mice"

_nutrients, 2023, doi:10.3390/nu15112580_

Round 1

Reviewer 1 Report

The manuscript titled “Effects of royal jelly on gut dysbiosis and NAFLD in db/db mice” investigated the effect of royal jelly on gut dysbiosis and NAFLD in vivo and in vitro. Authors provided a lot of evidence to demonstrate its effect. The authors have given some interesting results and some major revisions should be modified as follows:

1.      Only males are used in this study. Please include the rationale or discuss potential sex differences.

2.      A detailed description of the mice's diet is lacking. This information is required.

3.      In methodology, the replicas of mice are described clearly. In the Fig. 1, six replicas are presented for each treatment. However, in the Fig. 1, just three replicas are presented for gut microbiota, please explain it and describe the graph in detail for a better understanding.

4.      Lane 125-126, “16. S rRNA sequencing” should be “16 S rRNA sequencing”.

5.      The method of histological study and gut microbiota can refer to Doi.org/10.1039/D2FO03019E and DOI: 10.1039/d0fo03471a.

6.      The section of Conclusion was too short. Authors should write the main conclusion of this manuscript. Moreover, the defects of this paper in the next step can be reflected in the conclusion.

Reviewer 2 Report

Dear Authors, thank you for the interesting manuscript, but considering the title of the manuscript, which is supposed to focus on the analysis of changes in the gut microbiome under the influence of royal jelly, it is necessary to modify many things:

1. Please specify the purpose of the study, because currently the content of the manuscript shows that the focus is on MCFAs that affect many things and not on dysbiosis (i.e. disturbance of the intestinal microbiome)

2. Please specify for what purpose the particular test methods were used (e.g. blood glucose level), because the results were not correlated with the profile of intestinal bacteria and they were not analyzed for dysbiosis.

3. In the assessment of the gut microbiome, it is very important to describe the research methods (in this case, I can guess that it is about next-generation sequencing-NGS). The Authors give only a reference to their earlier publication (item no. 21 in References). Unfortunately, in that publication, the Authors also refer to supplementary materials, but I could not find a description of the method in them. Please, supplement the description with a reliable method of assessing the gut microbiome.

4. The results may not be complete. What role does the control group play? At what taxonomic levels has the gut microbiome been analyzed? Did you manage to get down to L7 (species)? How did bacterial profiles change at individual taxonomic levels depending on the concentration of royal jelly?

5. Did the Authors only analyze the 2 alpha diversity indices? Did the Authors also evaluate beta diversity? What conclusions can be drawn from the assessment of the biodiversity of the gut microbiome of the animals subjected to the experiment?

6. Notations on the figures and captions under the figures are not clear. Differences between which groups are statistically significant marked with asterisks? What do "n=6" and "n=3" mean? Is it the number of animals in each study group? If so, what is the statistical value of microbiome analyzes from only 3 animals?

7. Please check again: the content and vocabulary ("microorganism" sounds better than "microbe"), the order and correctness of the explanation of abbreviations (e.g. MCFAs p.2, NASH and FFAR p.14), the correctness of disease names (currently "diabetes" and not " diabetes mellitus"), the correct spelling of bacterial taxa (in italics only in genus and species names, "Bacteroides" is the name of the genus, not of the phylum, which is "Bacteroidetes")
